# Risk Factors for Recanalization after Coil Embolization

**DOI:** 10.3390/jpm11080793

**Published:** 2021-08-14

**Authors:** Karol Wiśniewski, Zbigniew Tyfa, Bartłomiej Tomasik, Piotr Reorowicz, Ernest J. Bobeff, Bartłomiej J. Posmyk, Marlena Hupało, Ludomir Stefańczyk, Krzysztof Jóźwik, Dariusz J. Jaskólski

**Affiliations:** 1Department of Neurosurgery and Neurooncology, Medical University of Lodz, Kopcińskiego 22, 90-153 Lodz, Poland; ernestbobeff@gmail.com (E.J.B.); posmyk.bartlomiej@gmail.com (B.J.P.); marhup2@gazeta.pl (M.H.); dariusz.jaskolski@umed.lodz.pl (D.J.J.); 2Institute of Turbomachinery, Medical Apparatus Division, Lodz University of Technology, Wolczanska 219/223, 90-924 Lodz, Poland; zbigniew.tyfa@dokt.p.lodz.pl (Z.T.); piotr.reorowicz@p.lodz.pl (P.R.); krzysztof.jozwik@p.lodz.pl (K.J.); 3Department of Biostatistics and Translational Medicine, Medical University of Lodz, 15 Mazowiecka St., 92-215 Lodz, Poland; bartektomasik@gmail.com; 4Department of Radiation Oncology, Dana-Farber Cancer Institute, Harvard Medical School, Boston, MA 02215, USA; 5Department of Radiology-Diagnostic Imaging, Medical University of Lodz, Kopcińskiego 22, 90-153 Lodz, Poland; ludomir.stefanczyk@umed.lodz.pl

**Keywords:** embolization, coils, recanalization, predictors, computational fluid dynamics, complex flow patterns, flow diverters

## Abstract

The aim of our study was to identify risk factors for recanalization 6 months after coil embolization using clinical data followed by computational fluid dynamics (CFD) analysis. Methods: Firstly, clinical data of 184 patients treated with coil embolization were analyzed retrospectively. Secondly, aneurysm models for high/low recanalization risk were generated based on ROC curves and their cut-off points. Afterward, CFD was utilized to validate the results. Results: In multivariable analysis, aneurysm filling during the first embolization was an independent risk factor whilst packing density was a protective factor of recanalization after 6 months in patients with aSAH. For patients with unruptured aneurysms, packing density was found to be a protective factor whilst the aneurysm neck size was an independent risk factor. Complex flow pattern and multiple vortices were associated with aneurysm shape and were characteristic of the high recanalization risk group. Conclusions: Statistical analysis suggested that there are various factors influencing recanalization risk. Once certain values of morphometric parameters are exceeded, a complex flow with numerous vortices occurs. This phenomenon was revealed due to CFD investigations that validated our statistical research. Thus, the complex flow pattern itself can be treated as a relevant recanalization predictor.

## 1. Introduction

Unruptured intracranial aneurysms occur in 2–5% of people [1], and a prognosis for subarachnoid bleeding from an aneurysmal subarachnoid hemorrhage (aSAH) is serious, as mortality is 60% within 6 months [2]. Coil embolization is often used to treat intracranial aneurysms. Results of the ISAT study have strengthened the belief in the safety and efficacy of the embolization procedure—low morbidity and mortality of 4.1% are very promising results [3,4].

However, the benefits of embolization are overshadowed by the fact that this procedure does not effectively eliminate intracranial aneurysms. Thus, embolization needs control studies [5]. This is highly important, especially for young people who are supposed to enjoy many years of life [6,7].

On the one hand, embolization is a low-invasive method [8], but on the other hand, it involves a high risk of recanalization, ranging from 2.5 to 41% [6,9,10,11,12,13,14]. The recanalization rate varies between aneurysm sizes and in small aneurysms is considered to be low [14,15,16].

The risk of rerupture of embolized aneurysms is estimated to be 1.7% in the first year and 0.21% in the subsequent years [17]. According to the CARAT study, the risk of rerupture varies depending on the extent of the aneurysm being filled with spirals [18]. Thus, the rerupture risk for completely embolized aneurysms is 1.8%; for 91–99% embolization of aneurysm, the risk is 2.3%; for 70–90% embolization, it is 4.9%; however, if the aneurysm filling is less than 70%, it increases to 25% within 4 years after treatment [18].

Imperfections associated with embolization often require re-embolization or surgical clipping, as recanalization occurrence is a risk factor for their rupture in the late period—later than 30 days following the embolization procedure [19,20].

Our work consists of two main parts. In the first step, we performed a retrospective analysis of clinical data and morphometric parameters to establish their selected values that were associated with recanalization. In the second step, we performed a computational fluid dynamics (CFD) analysis based on our findings on the aneurysm morphometric parameters. We prepared digital models for high and low recanalization risk and visualized specific flow patterns associated with aneurysm shape. To find differences between both groups, we examined flow patterns and hemodynamic parameters in all generated models. Such investigations are relevant as compaction relates to flow patterns [21]. Thus, it can be stated that CFD served as a tool that allowed us to cover two main aspects: (1) visualizing flow hemodynamics in the vicinity of all aneurysm models and performing a qualitative and quantitative comparison of obtained flow solution data and (2) validating whether the morphometric parameters associated with recanalization, indicated by our statistical analysis, result in noticeable changes in flow hemodynamics. This might include highly elevated wall shear stress (WSS) or oscillatory shear index (OSI), as well as a formation of recirculation zones in the aneurysm dome.

In our work, we merged the results from clinical and CFD analyses in order to characterize predictors of the recanalization. Hence, the goal of this study was to identify recanalization risk factors and evaluate their mechanistic basis. 

## 2. Materials and Methods

### 2.1. Ethics Approval

The study was approved by the local Bioethical Committee at the Medical University of Lodz, application number RNN/119/15/KE. All experiments were performed in accordance with relevant guidelines and regulations. The study was designed in accordance with the Good Clinical Practice (GCP) guidelines and was conducted according to the principles of the Declaration of Helsinki. Informed consent was obtained from the participants prior to inclusion. In case of depressed level of consciousness, the patient’s legal representative was asked for informed consent.

### 2.2. Patients

We conducted a retrospective analysis of digital subtraction angiograms (DSAs), computed tomography angiograms (CTAs) and patient notes of 410 cases of intracranial aneurysms treated with coil embolization over an 8-year period, and based on our statistical results, using computational fluid dynamics (CFD), we built models for high and low risk of recanalization to find new potential markers of recanalization risk. All patients had a 6-month follow-up DSA (according to our protocol and guidelines, the control DSA was done 6, 12 and 24 months after the first embolization).

In each case, CTAs were done and three-dimensional (3D) models were generated. The inclusion and exclusion criteria are listed in Table 1.

### 2.3. Morphometric Parameters

We performed the analysis of morphometric parameters based on 3D aneurysm models generated from preoperative CTA (all 3D CTAs were performed on GE VCT LightSpeed VCT with slice thickness of 0.625 mm and increment of 0.5 mm using 3D software). We measured the following: Aneurysm dome size (height, width, depth);The maximal perpendicular height;Neck size;Parent artery size;Aneurysm angle;Vessel angle;Neck to parent artery ratio;Aneurysm depth to neck size ratio;Aspect ratio (AR), defined as the maximal perpendicular height (the largest perpendicular distance from the neck of the aneurysm to the dome of the aneurysm) divided by neck width;Size ratio (SR), defined as maximum aneurysm height (between the center of the aneurysm neck and the greatest distance to the aneurysm dome) divided by vessel diameter [22]. (see Figure 1).

### 2.4. Angiography/Embolization

The measurements of the aneurysm size, the ratio of its sac to the neck and anatomical relations with the adjacent vessels were made with a computer after the previous calibration of the catheter located in the carotid or vertebral artery, on the base of digital images of subtractive angiography obtained in the anteroposterior and lateral projections with unified enlargement up to 22 cm.

Platinum embolization spirals (hydraulically or mechanically released) or electrolytically detachable spirals coated with hydrogel were used. Their diameters were 0.010″ and 0.018″ and they were available in two forms of coil: spatial (3D) and helical.

#### 2.4.1. Prevention of Thromboembolic Events

In the time of endovascular treatment of the aneurysm patients receive 5000 IU of unfractionated heparin intravenously, followed by 2000 IU for each hour of the procedure. The distal end of the Chaperon guide catheter (5F or 6F—depending on the diameter of the vessel) is placed in the first (carotid—C1) segment of the internal carotid artery (in the case of the vertebral artery, in segment V2). Then, 0.9% saline solution with heparin at 2000 IU/L is used to flush the guide catheter. This is considered a standard protocol for preventing procedural thromboembolic risks.

The use of stent-assisted coiling can reduce the recanalization rate after coil embolization [23,24]; thus, the use of stents became widespread, and the need for preventive antiplatelets became necessary before the treatment of unruptured aneurysms [25,26]. Current guidelines recommend a 100–300 mg/day dosage of aspirin and a 75 mg/day dosage of clopidogrel (which inhibits the P2Y12 adenosine diphosphate receptor on the platelet cell membrane) administered orally 5–7 days before the stent-assisted coiling procedure. If a patient did not follow such a regimen, a loading dose of 300–500 mg of aspirin and 300 mg of clopidogrel can be administered just before coiling, as a viable alternative [27,28]. 

Clopidogrel is a standard medication for preventing thromboembolic events. However, the efficacy of prophylactic CPG therapy differs among the patients. Resistance to clopidogrel could occur [29,30,31,32]. Various platelet function tests can be used to identify the resistance. Several point-of-care devices are available to measure platelet functions. If resistance to clopidogrel is present, one of the other drugs such as prasugrel, cilostazol or ticlopidine should be added [33]. 

Oral prophylaxis of thromboembolism is widely accepted in unruptured cerebral aneurysms. The use of antiplatelet therapy in ruptured aneurysms is unclear and needs more study [34,35]. To prevent thromboembolisms in ruptured intracranial aneurysm embolization, glycoprotein (Gp) IIb/IIIa inhibitors (abciximab, tirofiban or eptifibatide) could be used [36,37,38,39]. The mechanisms of action of Gp IIb/IIIa inhibitors include prevention of platelet aggregation, leading to reduction both in thrombus mass and the platform for further thrombin generation; inhibition of release of local thrombolysis inhibitors by platelets; and weakening of the clot structure by decreasing the binding of factor XIII and fibrin and α-2 plasminogen inhibition [40].

Tirofiban, is a nonpeptide tyrosine derivative that inhibits fibrinogen-dependent platelet aggregation in a dose-dependent manner, and its effect is reversed within 3 h of infusion [41], whereas abciximab is a humanized hybrid monoclonal antibody that binds to Gp IIb/IIIa and has a longer reversal time (12–24 h) compared with other glycoprotein IIb/IIIa (GP IIb/IIIa) inhibitors that may be undesirable in patients harboring ruptured cerebral aneurysms [42]. Eptifibatide has lower receptor affinity, higher plasma levels and is shorter-acting—2–4 h [43]. Long plasma half-life and low plasma concentrations make it difficult to reverse bleeding complications with platelet transfusions; thus, short-acting drugs are preferred in ruptured aneurysm embolization [44].

#### 2.4.2. Assessment of Aneurysm Volume, Packing Density, Recanalization and Degree of Aneurysm Filling during the First Embolization

The volume of the aneurysm and the presence of recanalization were determined on the base of DSA images. The diameter of the aneurysm meant its largest diameter, expressed in millimeters. The volume of the aneurysm was calculated using the following formulas tailored to its shape (spherical, ellipsoid or bilobed):

Spherical shape:$$\mathrm{volume}=\frac{\pi \;\cdot {\;\mathrm{diameter}}^{3}}{6}$$

Ellipsoid shape:$$\mathrm{volume}=\frac{\pi \;\cdot {\;\mathrm{diameter}}^{2}\;\cdot \mathrm{height}}{6}$$

Bilobed shape: ellipsoid A + ellipsoid B 

The spiral volume was calculated according to the following formula:$$\mathrm{volume}=\frac{\pi \;\cdot {\;\mathrm{diameter}}^{2}}{4}\;\cdot \mathrm{length}$$

The packing density was calculated according to the following formula:$$\mathrm{Packing}\;\mathrm{density}=\frac{\mathrm{spiral}\;\mathrm{volume}\;}{\mathrm{aneurysm}\;\mathrm{volume}}\;\cdot 100\%$$

Two spiral diameters were taken into account: 0.010″ and 0.018″. The volume of the aneurysm was calculated in cubic millimeters and the packing density was calculated in percentage.

The effectiveness of endovascular treatment was assessed visually after the embolization and at the follow-up DSA after 6 months. The modified Raymond-Roy scale was used to assess recanalization. Recanalization was diagnosed if there was increased aneurysm filling compared to the examination performed immediately after the embolization.

All radiological findings were measured independently by a radiologist and a neurosurgeon. Afterward, the average value was calculated.

### 2.5. Statistical Analysis 

Qualitative variables were compared using the chi-square test. Continuous variables were analyzed using *t*-test or Mann-Whitney U test, depending on the data distribution. Correlations were assessed using Spearman’s rank correlation. Receiver operating characteristic (ROC) curves were used to assess the quality of the classifiers, and the best cut-off point was determined using the Youden method. For multivariable analysis, we chose items that were significant in univariate analysis (after Bonferroni correction for multiple comparisons). Stepwise logistic regression with backward feature elimination was implemented for feature selection. The statistical analysis was performed using the Statistica 13.1 package (Statsoft 13.1, Cracow, Poland). *p* values < 0.05 were considered statistically significant.

### 2.6. Computational Fluid Dynamics Analysis

Since CFD provides data on the investigated flow with high spatial and temporal resolution that cannot be obtained during current clinical trials and diagnostics, it was decided to prove that numerical simulations of the blood flow could serve as a source of information (a predictor) for early recanalization. Therefore, models for two groups (high and low recanalization risk group) according to the optimal cut-off points for significant values, determined using the Youden method, were generated. Afterward, transient numerical simulations, with pulsatile flow conditions, were carried out. Apart from analyzing the flow patterns within the parent vessel and aneurysm dome (to assess the flow vorticity), we investigated other hemodynamic parameters, such as wall shear stress (WSS) at the given time step, time-averaged wall shear stress (TAWSS) and oscillating shear index (OSI). WSS is referred to as a tangential force of the flowing blood that acts upon the blood vessel wall, resulting in the deformation of the cells along the flow direction. Contrary to WSS, which is measured at the exact point in time, TAWSS is calculated as an arithmetic average WSS taken from the given time interval, e.g., the entire cardiac cycle. The last parameter, OSI, is known as temporal fluctuation of low and high average shear stresses, which helps in determining whether the WSS vector is aligned with the TAWSS one over the full cardiac cycle.

#### 2.6.1. Model Geometry Preparation

To identify a recanalization marker and to evaluate the influence of the aneurysm and its morphological parameters on the flow hemodynamics within the investigated domain, several geometries were created with the use of SolidWorks software. Each model was based on the referential (parent) case which was parametrized. This means that this parent geometry was prepared in such a way that the authors could change the desired morphological parameter, i.e., parent vessel diameter (P), aneurysm neck width (N), aneurysm height (H) and aneurysm max. width (S). Afterward, the program automatically updated the geometry based on new parameters. Therefore, the spatial alignment and overall shape of the parent vessel, aneurysm and its bifurcation angle were maintained in each analyzed case. The only parameters that changed were those of utmost importance, i.e., morphological parameters described earlier in this study. Due to such an approach, the authors could analyze changes in flow hemodynamics resulting only from the differences in desired parameters (for more details, see Appendix A). 

#### 2.6.2. 3D CFD Model Building—Other Possibilities

We think that incorporation of 2D DSA during 3D model preparation for CFD purposes is limited to the spatial size of the given image set. To reliably recreate a patient-specific geometry from biomedical imaging, not only should the resolution of the images be high, but also a significant number of subsequent slices (in all three dimensions) is required. Unfortunately, standard 2D DSA lacks information on a dense image set packing across all three dimensions. Thus, reconstruction of 3D geometry from such an insufficient dataset could be problematic, or the output model would not mimic the real patient-specific model due to several assumptions.

3D DSA followed by volume rendering (VR) method seems to be a more prominent approach when it comes to a possible 3D CFD model reconstruction. It is claimed that such a technique can more accurately depict the direction of the aneurysm neck, preserving all the angles, and allows avoidance of vessels overlapping. Due to volume determination, instead of using a specific reconstruction threshold (e.g., in MIP), 3D DSA is not influenced by thresholds. When compared to standard CTA, 3D DSA is not limited due to axial source section and it does not overproject hard tissues, so vessels are more distinguishable from the surrounding bone structures. However, as in a majority of diagnostic systems devoted to 3D model preparation, all the generated models serve only for visualization purposes. There is a lack of possibility of exporting such a rendered volume as a 3D model compatible with CFD software. Therefore, additional programs are required. Additionally, DSA systems are characterized by slightly different encoding protocols; thus, despite providing DICOM files, such images cannot be opened with the use of typical DICOM decoding libraries. This makes it even harder to find software that can decompress/decode such images and simultaneously generate 3D models suitable for CFD purposes.

#### 2.6.3. Numerical Simulations

Before conducting target numerical simulations, a mesh independence test was carried out to ensure the most optimal meshing parameters. The blood was modeled as non-Newtonian, incompressible fluid characterized by a power law viscosity model. Since the authors assumed no heat exchange, the simulated flow was set as an adiabatic and isothermal one. Numerical analyses were performed in Ansys CFX solver as transient (with pulsatile boundary conditions) and stationary simulations, where the latter served as initial conditions for the former ones. It was decided to use a standard turbulence model for blood flow simulations, i.e., k-ω shear stress transport (SST). As far as transient analyses are concerned, they enabled simulating four full cardiac cycles with a time step equal to 0.01 s. Such a number of pulses ensured negligible variation of hemodynamic parameters between the latter heart cycles and independence from the initial boundary conditions. A more detailed description of the numerical simulations is provided in the Appendix A.

## 3. Results

### 3.1. Patients

A total of 410 patients were screened for eligibility, and 184 were included (study flowchart is presented in Figure 2). The majority of bleeding aneurysms were coiled within 72 h after the SAH—48 cases. We did not find significant associations between clinical factors (hypertension, diabetes and smoking) and recanalization. In the overall group, the odds ratios (ORs) were as follows: OR = 0.99 (95%CI: 0.52–1.89), *p* = 0.970 for HA; OR = 0.50 (95%CI: 0.11–2.37), *p* = 0.385 for DM; and OR = 0.64 (95%CI: 0.33–1.24), *p* = 0.188 for smoking. In the group with ruptured aneurysms, the ORs were as follows: OR = 0.45 (95%CI: 0.14–1.48), *p* = 0.189 for HA; OR = 0.88 (95%CI: 0.09–9.08), *p* = 0.916 for DM; and OR = 0.49 (95%CI: 0.16–1.50), *p* = 0.213 for smoking. In the group with unruptured aneurysms, the ORs were as follows: OR = 1.46 (95%CI: 0.65–3.28), *p* = 0.364 for HA; OR = 0.35 (95%CI: 0.04–2.95), *p* = 0.333 for DM; and OR = 0.75 (95%CI: 0.33–1.69), *p* = 0.487 for smoking.

### 3.2. Laboratory Results, Morphometric Parameters, Aneurysm Volume, Packing Density of Intracranial Aneurysms and Complete Aneurysm Filling during the First Embolization

In the univariate analyses comparing patients with and without recanalization and performed in the group of patients with unruptured aneurysms, we observed significant differences (after Bonferroni correction for multiple hypotheses testing) in aneurysm depth (12.0 ± 6.6 mm vs. 6.7 ± 3.6 mm, *p* < 0.001), aneurysm height (13.8 ± 6.6 mm vs. 8.2 ± 4.7 mm, *p* < 0.001), aneurysm width (12.1 ± 6.8 mm vs. 6.6 ± 3.8 mm, *p* < 0.001), aneurysm neck size (5.0 ± 2.3 mm vs. 3.6 ± 1.0 mm, *p* < 0.001), aneurysm volume (1794.1 ± 2202.0 mm^3^ vs. 426.7 ± 1287.4 mm^3^, *p* < 0.001), packing density (20.0 ± 8.9% vs. 33.3 ± 8.0%, *p* < 0.001), SR ratio (3.4 ± 1.6 vs. 2.3 ± 1.3, *p* < 0.001), maximal perpendicular height (12.9 ± 6.0 mm vs. 7.7 ± 4.3 mm, *p* < 0.001) and maximal perpendicular height/neck size (aspect ratio) (12.7 ± 6.0 vs. 7.8 ± 4.6, *p* < 0.001). 

In the univariate analyses comparing patients with and without recanalization and performed in the group of patients with ruptured aneurysms, we observed significant differences (after Bonferroni correction for multiple hypotheses testing) in aneurysm neck size (4.1 ± 0.7 mm vs. 3.3 ± 0.8 mm, *p* = 0.001), packing density (21.2 ± 6.6% vs. 35.0 ± 10.8%, *p* = 0.001) and maximal perpendicular height (12.0 ± 5.1 mm vs. 6.9 ± 3.4 mm, *p* = 0.001). Detailed results of all performed analyses are presented in Table 2.

### 3.3. Type of Embolization Material 

In the group of aSAH patients, platinum spirals were used in 64 cases, whereas bioactive spirals were used only in 16 cases. With regard to the unruptured group, platinum spirals were used in 111 patients, and bioactive spirals were used in 102 patients.

### 3.4. Aneurysm Location 

We noted that the majority of ruptured aneurysms were located in the middle cerebral artery and that the majority of aneurysms in the unruptured group were located in the internal carotid artery. Detailed results are presented in Table 3. We did not find any association between aneurysm location and recanalization in both groups.

### 3.5. Recanalization (after 6 and 12 Months)

In the group with unruptured aneurysms, recanalization after 6 months was found in 33 cases (28%). In 14 cases, it was class IIIa; in 15 cases, it was class IIIb; and in 4 cases, it was class II, according to the modified Raymond–Roy scale. Repeat treatment was done in 29 patients.

In the SAH group, recanalization was found in 18 cases (27%); in 12 cases, it was class IIIb according to the modified Raymond–Roy scale, and in 6 cases, it was class IIIa. The treatment was repeated in 18 patients. 

After 12 months, recanalization was observed in 24 cases in the unruptured group (20%), and the treatment was repeated in 14 cases (11.8%). In the ruptured group, recanalization was noted in 12 cases (18%), and the treatment was repeated in 8 cases (12.1%).

In our sample, the highest retreatment rate was observed after 6 months in both groups, confirming the importance of the first DSA after 6 months.

### 3.6. Unruptured Aneurysm—Statistical Analysis

#### 3.6.1. Early Recanalization (after 6 Months) in the Unruptured Aneurysm

Of all patients included in the study, 64.1% (*n* = 118) had an unruptured aneurysm. Here, recanalization was found in 33 cases (28%) after 6 months. The results of the univariate analysis are presented in Table 2.

Univariate analyses showed factors favoring recanalization, whereas ROC curves enabled the creation of a differential classifier between the groups at low and high risk of aneurysm recanalization. The factors that were juxtaposed with the optimal cut-off points increased the likelihood of early recanalization.

In the unruptured aneurysm group, recanalization would occur if the following conditions were met (according to ROC curve analysis):Aneurysm width >9.3 mm (AUC (95%CI) 0.76 (0.64–0.87); *p* < 0.001);Aneurysm height >13 mm (AUC (95%CI) 0.76 (0.64–0.87); *p* < 0.001);Aneurysm depth >11 mm (AUC (95%CI) 0.73 (0.60–0.85); *p* < 0.001);Aneurysm neck width >4 mm (AUC (95%CI) 0.82 (0.73–0.90); *p* < 0.001);Diameter of the parent artery >4.6 mm (AUC (95%CI) 0.63 (0.51–0.75); *p* = 0.029);SR ratio >2.759 (AUC (95%CI) 0.72 (0.61–0.84); *p* < 0.001);Index determining the ratio of neck width to diameter of the parent artery >1.042 (AUC (95%CI) 0.66 (0.55–0.76); *p* = 0.004);Largest aneurysm dimension perpendicular to the neck >12.7 (AUC (95%CI) 0.76 (0.65–0.87); *p* < 0.001);Index determining the ratio of the largest aneurysm dimension perpendicular to the neck to the width of the aneurysm neck (aspect ratio) >10.526 (AUC (95%CI) 0.76 (0.64–0.87); *p* < 0.001);AR ratio >10 (AUC (95%CI) 0.72 (0.60–0.84); *p* < 0.001);Packing density <23.5% (AUC (95%CI) 0.86 (0.77–0.95); *p* < 0.001);INR on embolization day >1.01 (AUC (95%CI) 0.62 (0.51–0.73); *p* = 0.036);Prothrombin index on embolization day >99.8 (AUC (95%CI) 0.62 (0.51–0.73); *p* = 0.036).

#### 3.6.2. Assessment of Recanalization Depending on the Degree of Aneurysm Filling during the First Embolization

In the unruptured aneurysm group, complete aneurysm filling during the first embolization significantly reduced the probability of recanalization—odds ratio (OR) 0.014, 95%CI: 0.001–0.25, *p* < 0.001.

#### 3.6.3. Assessment of Recanalization Depending on the Type of Applied Embolization Spirals 

In the unruptured aneurysm group, the use of platinum spirals significantly reduced the likelihood of recanalization—OR 0.13, 95%CI: 0.02–0.73, *p* = 0.018.

### 3.7. Ruptured Aneurysm—Statistical Analysis

#### 3.7.1. Early Recanalization of Ruptured Aneurysms

Patients with ruptured aneurysm accounted for 35.9% (*n* = 66). Here, recanalization was found after 6 months in 18 cases (27.3%). The results of the univariate analysis are presented in Table 2.

Univariate analyses showed factors favoring recanalization, and ROC curves enabled the creation of a differential classifier between the groups at low and high risk of aneurysm recanalization. The factors that were juxtaposed with the optimal cut-off points increased the likelihood of early recanalization.

In the aSAH group, recanalization would occur if the following conditions were met (according to ROC curve analysis):Aneurysm height >12 mm (AUC (95%CI) 0.65 (0.49–0.81); *p* = 0.063);Aneurysm neck width >3.6 mm (AUC (95%CI) 0.83 (0.72–0.95); *p* < 0.001);Packing density <27.5% (AUC (95%CI) 0.88 (0.79–0.98); *p* < 0.001);Index determining the ratio of neck width to diameter of the parent artery >1.023 (AUC (95%CI) 0.67 (0.52–0.82); *p* = 0.025);Largest aneurysm dimension perpendicular to the neck >12 mm (AUC (95%CI) 0.77 (0.62–0.92); *p* < 0.001);Index determining the ratio of the largest dimension of the aneurysm perpendicular to the neck to the width of the aneurysm neck (aspect ratio) >3.075 (AUC (95%CI) 0.68 (0.51–0.85); *p* = 0.039).

#### 3.7.2. Assessment of Recanalization Depending on Aneurysm Filling during the First Embolization

Complete aneurysm filling significantly reduced the likelihood of recanalization rate—OR 0.02, 95%CI: 0.004–0.13, *p* < 0.001.

#### 3.7.3. Assessment of Recanalization Depending on Coil Types 

In the aSAH group, the application of bioactive spirals significantly increased the likelihood of recanalization—OR 8.31, 95%CI: 1.01–68.34, *p* = 0.049.

### 3.8. Multifactorial Analysis for Ruptured and Unruptured Aneurysms

In the multivariable analysis, we showed that for patients with aSAH, the degree of aneurysm filling during the first embolization was an independent risk factor (OR 17.00 (95%CI: 2.32–124.40)) while packing density was a protective factor (OR 0.87 (95%CI: 0.79–0.97)) of recanalization after 6 months. For patients with unruptured aneurysms, we identified that packing density was again a protective factor (OR 0.84 (95%CI: 0.79–0.91) while the aneurysm neck size was an independent risk factor (OR 2.16 (95%CI: 1.23–3.80)) of recanalization after 6 months. Next, we checked whether digital models generated using the same set of variables could predict the risk of recanalization after 12 months; according to the literature, DSA after 12 months can be used as a first control [16]. We found that these sets of variables could robustly predict the risk of recanalization after one year. The detailed results are presented in Table 4 and Figure 3 demonstrating ROC curves for each model.

### 3.9. Computational Fluid Dynamics Analysis

As previously stated, we generated several geometries to identify recanalization markers and to support the performed statistical analysis via numerical predictions. After estimating the values that increase the recanalization risk (based on ROC curve analysis), we chose 14 configurations of recanalization markers for the unruptured aneurysm group and 8 configurations for the ruptured aneurysm group.

Additionally, we conducted two reference case studies, one for the ruptured aneurysm group and one for the unruptured group. Reference geometries were prepared based on the statistical results presented in Section 3.6.1 and Section 3.7.1. The morphologic parameters used for each case are gathered in the Appendix A. The analysis of the results focused on an examination of numerous hemodynamic parameters, including wall shear stress (WSS), time-averaged wall shear stress (TAWSS), vorticity, flow structure and oscillatory shear index (OSI). It is worth mentioning that all those investigations were carried out only for the aneurysm vicinity to avoid the influence of low WSS values of the parent and child vessels.

In the high recanalization risk group, we noted low maximal WSS and TAWSS values, complex flow patterns and multiple vortices in all the models. Greater aneurysm size was associated with lower maximal WSS and TAWSS values and a more complex flow pattern. In the low recanalization risk group, in all the models, the results were opposite (high maximal WSS and TAWSS values and simple flow patterns). For instance, for case #2 in the unruptured aneurysm group, maximal values of WSS for high and low recanalization risk geometries were equal to 90 and 96 Pa, respectively (circa 7% difference). For the same case, but in the ruptured aneurysm group, maximal values of WSS were equal to 102 and 132 Pa for high and low recanalization risk geometries, respectively. As can be seen, this difference is significant, i.e., 30 Pa (29%). However, it has to be noted that the maximal value of WSS is a local parameter—it outlines the maximal value of WSS at a specific node of the mesh. 

As far as OSI is concerned, similarly to the area-averaged WSS, it is a global parameter. Based on all the calculated data, we could not find any correlation between OSI and recanalization in all the models built both for high and low risk of recanalization (Table 5, Figure 4, Figure 5 and Figure 6, Appendix A).

As far as velocity analysis is concerned, a qualitative inspection of flow patterns and vortices could be performed by investigating prepared animations. Unfortunately, presented images taken at a specific timestep (Figure 4, Figure 5 and Figure 6) might not present a phenomenon of vortices generation and their propagation—they depict flow patterns at the chosen time value. Nonetheless, it could be concluded that a complex flow pattern and multiple vortices were characteristic only of morphometric parameters that favor recanalization, i.e., in the high recanalization risk group. Thus, when the aneurysm morphometric parameters reach certain dimensions, a complex flow pattern and multiple vortices appear.

Furthermore, it was decided to investigate how velocity changes at the aneurysm neck cross-section in each analyzed case study. Table 6 comprises numerical results of such research. As can be seen, a vast majority of low recanalization risk case studies (unruptured group) are characterized by slightly higher relative differences, with a reference model, than models in the high recanalization risk group. Lower blood inflow to the aneurysm dome might mean that low recanalization risk models exhibit topology that makes it harder for blood to flow into the aneurysm and, consequently, might not result in a future recanalization. The most significant changes could be noted for case study #3 (N > 4 mm) and case study #6 (N/P ratio > 1.042), where the relative differences between high and low recanalization risk groups exceeded 30%. One could not perform similar observations for the ruptured group—hardly any correlation between the morphometric parameters differences and blood velocity changes could be found.

## 4. Discussion

Dynamic technological progress leads to the emergence of new solutions before we can fully appreciate the safety and effectiveness of those already accepted [45,46]. This fact hinders the homogeneous assessment of selected endovascular treatments.

In our material, the highest retreatment rate was observed after 6 months. For this reason, we assessed the recanalization predictors and CFD analysis concerning this period. According to the literature, 46.9% of all recanalizations occur within 6 months after the procedure, with nearly 40% being of major degree [47,48]. In a group of 690 coiled aneurysms, Jeon et al. noted that early recanalization occurred more often than the late one (128 vs. 45) [12].

It is already known that large, wide-necked ruptured aneurysms and those with atherosclerotic plaques often recanalize [48,49,50,51]. The literature also states that incomplete primary embolization [52,53,54] and packing density [55,56,57,58,59] are associated with recanalization (Table 7). This was also confirmed in our analysis.

Additionally, we found a correlation between the INR and prothrombin index and recanalization in the unruptured aneurysm group. Only when the blood’s ability to clot is normal, the pressure in the sac may be reduced significantly enough to contribute to the therapeutic success. Thus, normal clotting plays a key role in reducing aneurysmal sac pressure and reducing the risk of aneurysm rupture after coiling [66]. Insufficient coagulation probably promotes the spiral compaction and the following recanalization, but this hypothesis needs to be verified.

An up-to-date literature review shows only a few reports that describe some morphometric parameters affecting recanalization, none of which have confirmed statistical results experimentally, e.g., using CFD. Thus, there is a lack of reliable research specifying the criteria for specific endovascular procedures.

Our CFD analysis revealed different flow patterns depending on the aneurysm morphometric parameters. Thus, a complex flow pattern and multiple vortices were characteristic only for morphometric parameters that favored recanalization—such flow pattern could be observed in all the CFD models in the high recanalization group. We believe that morphometric parameters prompt recanalization by being the cause of the complex flow pattern in the aneurysmal sac. In this study, we highlighted that a complex flow pattern and multiple vortices could play a decisive role in recanalization and therefore allow it to be predicted. It was established that they are responsible for compaction of the coil mesh (the decrease in interspaces between the loops of the coils, which results in a smaller coil mesh) and lead to reopening of the aneurysmal lumen.

We emphasize that when the aneurysm morphometric parameters reach certain values, the complex flow pattern and multiple vortices appear and become the direct cause of recanalization. This hypothesis needs to be verified in future studies in which we plan to prospectively demonstrate that the complex flow pattern, if shown in virtual modeling before endovascular treatment, foretells a greater risk of recanalization. From this perspective, this paper may be treated as an interim study. The use of CFD analysis in clinical practice is a novel method with great potential. CFD provides a vast amount of data of high spatial and temporal resolution that cannot be obtained as a result of the clinical trials. Hence, verifying its validity on ‘artificial cases’ might indicate that CFD could be used preoperatively in patient-specific cases. Accurate vessel geometry can be obtained using high-resolution medical imaging modalities, and hemodynamic analysis can be performed by means of CFD tools that approximate flow solution during the numerical computation stage. With CFD analysis, one can assess hemodynamic parameters for predicting an aneurysm recanalization.

The conducted research shows that the CFD tool can be used to support physicians in diagnosing patients of different ages and various anatomical arterial structure anomalies and may help in clinical decision making. In our analysis, we noted that the complex flow pattern and multiple vortices constituted the essential difference. This value gives an insight into the blood flow inside the aneurysmal sac and could serve as a recanalization predictor. In the future, we plan to perform a prospective study, in which we will model the hemodynamics changes before and after the coiling procedure for all the aneurysm cases and then identify hemodynamics that correlates with the recurrence of coiled aneurysms.

It should be mentioned that in the studies that examined aneurysm vortices, complex flow patterns and multiple vortices were found to be associated with ruptured aneurysms [67,68]. Such complex flow patterns increase inflammatory cell infiltration in the aneurysm wall and could suggest a propensity to develop intrasaccular thrombus. In general, simple stable flow patterns with a large vortex are seen in stable aneurysms. In contrast, the complex unstable flow patterns in unstable, growing and ruptured aneurysms are more likely to result in multiple smaller vortices.

According to our results, a complex flow pattern, cerebral vessel turbulent blood flow (multiple vortices), occurs in the high recanalization risk group and may result in aneurysm recanalization following coiling. Turbulent blood flow promotes leaching of the thrombus, i.e., aneurysm recanalization despite any previous closure with coils [69]. This kind of flow might result in an increase in the mobility of the vibrating vessel wall and possibly rinsing out the thrombus from the aneurysm. Thus, in our opinion, intrasaccular flow pattern is an important recanalization predictor, which has been confirmed by our results.

WSS (as the tangential drag force per unit area of the endothelial surface) [70] was examined in a number of CFD studies. It is the most important CFD parameter [71]. In most studies, low WSS was associated with unstable aneurysms (high risk of rupture and growth). Concerning recanalization, the opinions are divided. In several studies, higher WSS was associated with aneurysmal recanalization [72,73]. In contrast, other studies showed that aneurysm recurrence was related to a low WSS, which may induce apoptosis of aneurysm endothelial cells [74]. Zhang et al. reported that a low amplitude of hemodynamic reduction ratio (RR) of WSS after embolization was observed in the recanalization group [75]. Our study emphasizes the role of low maximal wall shear stress (WSS) values in recanalization.

We have also shown that maximal TAWSS value was lower in the higher recanalization risk group in the ruptured and unruptured aneurysms models. This is probably because the larger the size of the aneurysm (the larger open space inside the aneurysm) and the lower the inflow rate, the lower the TAWSS will be. Currently, there are no data about the TAWSS and its affinity to recanalization in the literature. Fukuda et al. demonstrated that an unstable, recirculating flow structure within the aneurysm sac created in the region adjacent to the aneurysm wall with low TAWSS could be used to predict the risk of growth [76]. Thus, the results of this research indicate that low maximal TAWSS values might be one of the new predictors of recanalization.

We are aware that the maximal values of WSS and TAWSS are local parameters—they outline the maximal value of WSS and TAWSS at a specific node of the mesh, and one cannot draw clinical conclusions based on those values. However, in our opinion, intrasaccular flow pattern can be an important recanalization predictor. Once seen before the embolization, it promotes compaction which is responsible for aneurysm recanalization. Blood flow patterns influence compaction [21]. Our analysis showed that the complex flow patterns are closely related to the aneurysm shape.

Our work is the first such cross-sectional attempt to perform additional CFD analysis to provide mechanistic insights into the physical basis of our statistically observed clinical results.

Should there be considerable recanalization risk factors, it may be reasonable to consider other endovascular treatment. This would help to avoid the risk of a secondary intravascular procedure, which carries a higher complication rate than the first embolization procedure [77] and exposes patients to an additional dose of X-rays.

In our clinical practice, the results of the analysis lead to a better preparation for embolization. If the values of the predictors exceed the level of the cut-off points (the complex blood flow with multiple vortices has occurred), then a patient with an unruptured aneurysm is prepared for embolization with antiplatelet therapy (captopril 75 mg 1 × 1 p.o. 7 days before the procedure; in case of clopidogrel resistance, additional prasugrel is used: a 20 mg loading dose and a 5 mg maintenance dose). We do not use antiplatelet therapy in all patients with unruptured aneurysms regularly. This approach selects a group of patients in whom stent implantation is necessary to obliterate the aneurysm by stent-assisted embolization or flow diverter (FD) implantation. It protects patients from unnecessary stent implantation and the use of antiplatelet drugs, thus decreasing the risk of complications.

In patients with ruptured aneurysms, the predictors inform us that eptifibatide should be used intraoperatively (with a bolus dose of 180 μg/kg administered intravenously). Eptifibatide prevents thromboembolic complications during stent-assisted embolization and FD implantation. It does not increase the hemorrhagic risk. When eptifibatide is given in a bolus, patients are not maintained on continuous intravenous heparin infusion for the usual 12–24 h after the procedure. After placement of a stent-assisted coiling, patients are loaded with 300 mg of clopidogrel and then placed on 75 mg/day clopidogrel and 325 mg/day aspirin postprocedurally for 30 days and then 325 mg/day aspirin indefinitely.

The limitation of this study is conducting morphometric analyses on 3D CTA models, i.e., a test that requires contrast. The presence of contrast in a vessel lumen never perfectly reflects the conditions prevailing in its interior, and the creation of a 3D model of brain vessels depends on the radiologist analyzing the study. We conducted a morphometric analysis of images obtained from the CTA examination because it is currently the most accurate study of cerebral vessels. To reduce the margin of error, 3D aneurysm models were made independently by two radiologists. The misbalance between ruptured and unruptured groups exists because, at our department, ruptured aneurysms of the anterior circle of Willis are surgically occluded within 12 h after admission. As to CFD models’ limitations, they are based on many assumptions, such as rigid aneurysm walls, specific non-Newtonian flow and normal physiologic conditions. Thus, the coarse temporal resolution used in these simulations may mask flow instabilities.

## 5. Conclusions

Various factors influence the risk of recanalization. ROC curves and optimal cut-off points for morphometric parameters provide borders after which recanalization probability increases and coiling may be unsuccessful. Moreover, we draw attention to that when the aneurysm morphometric parameters reach certain dimensions, a complex flow pattern and multiple vortices appear. This phenomenon was revealed due to CFD investigations that validated our statistical research. Thus, intrasaccular flow pattern is responsible for coil mesh compaction and should be treated as a relevant recanalization predictor.

## Figures and Tables

**Figure 1 jpm-11-00793-f001:**
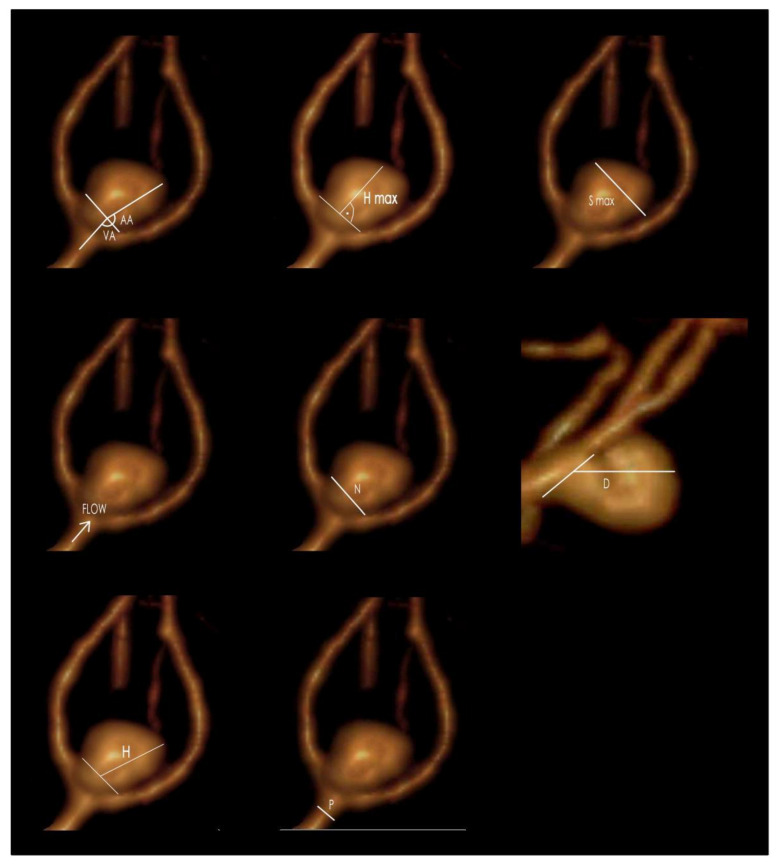
The morphometric parameters (flow—blood flow direction): S max—maximal aneurysm dome size; N—neck size; H—maximum aneurysm height, measured between the center of the aneurysm neck and the greatest distance to the aneurysm dome; H max—maximal perpendicular height, the largest perpendicular distance from the neck of the aneurysm to the dome of the aneurysm; P—parent artery diameter; AA—aneurysm angle; VA—vessel angle; D—maximum aneurysm depth (GIMP 2.10.22 DMG revision 3 for macOS).

**Figure 2 jpm-11-00793-f002:**
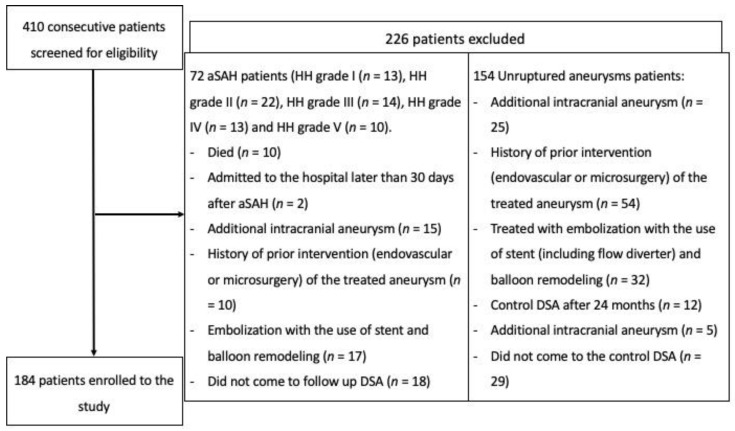
Flowchart presenting the enrollment of the patients.

**Figure 3 jpm-11-00793-f003:**
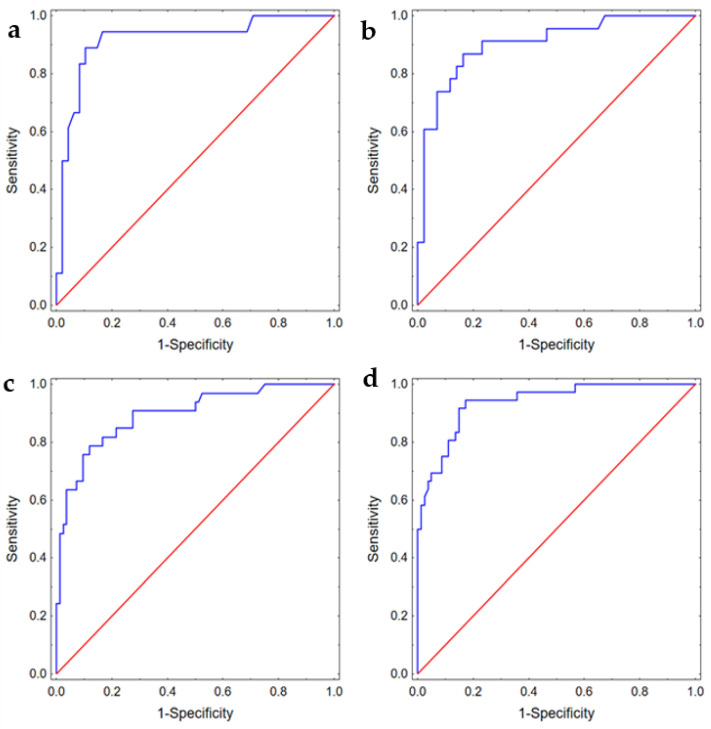
ROC curves presenting discriminating abilities of the models predicting (**a**) recanalization after 6 months in patients with aSAH, AUC 0.92 (95%CI: 0.83–1.00); (**b**) recanalization after 12 months in patients with aSAH, AUC 0.90 (95%CI: 0.83–0.98); (**c**) recanalization after 6 months in patients with unruptured aneurysm, AUC 0.90 (95%CI: 0.83–0.96); (**d**) recanalization after 12 months in patients with unruptured aneurysm, AUC 0.94 (95%CI: 0.89–0.98).

**Figure 4 jpm-11-00793-f004:**
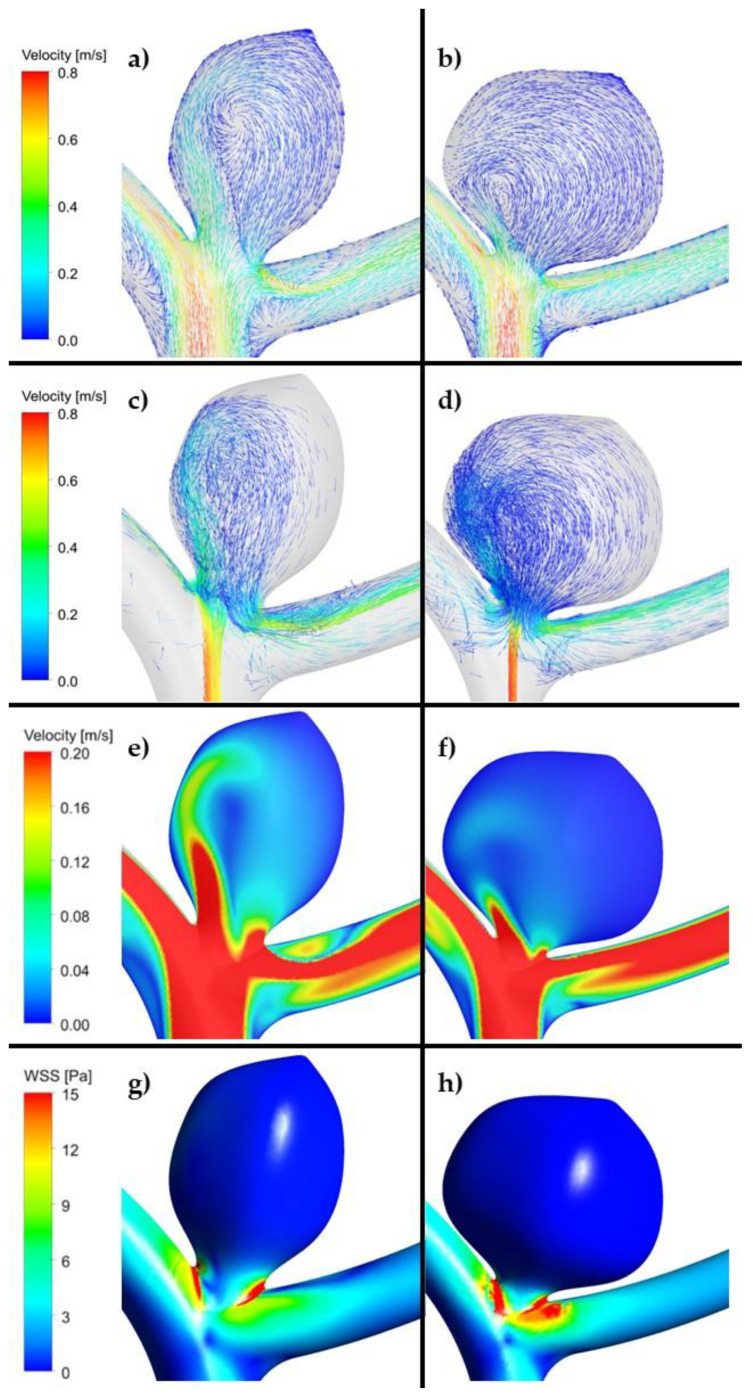
(**a**,**c**,**e**,**g**). Example of reference aneurysm model (unruptured group). (**b**,**d**,**f**,**h**). Example of reference aneurysm model (ruptured group). (ANSYS Inc. Post-processing module, Canonsburg, PA, USA).

**Figure 5 jpm-11-00793-f005:**
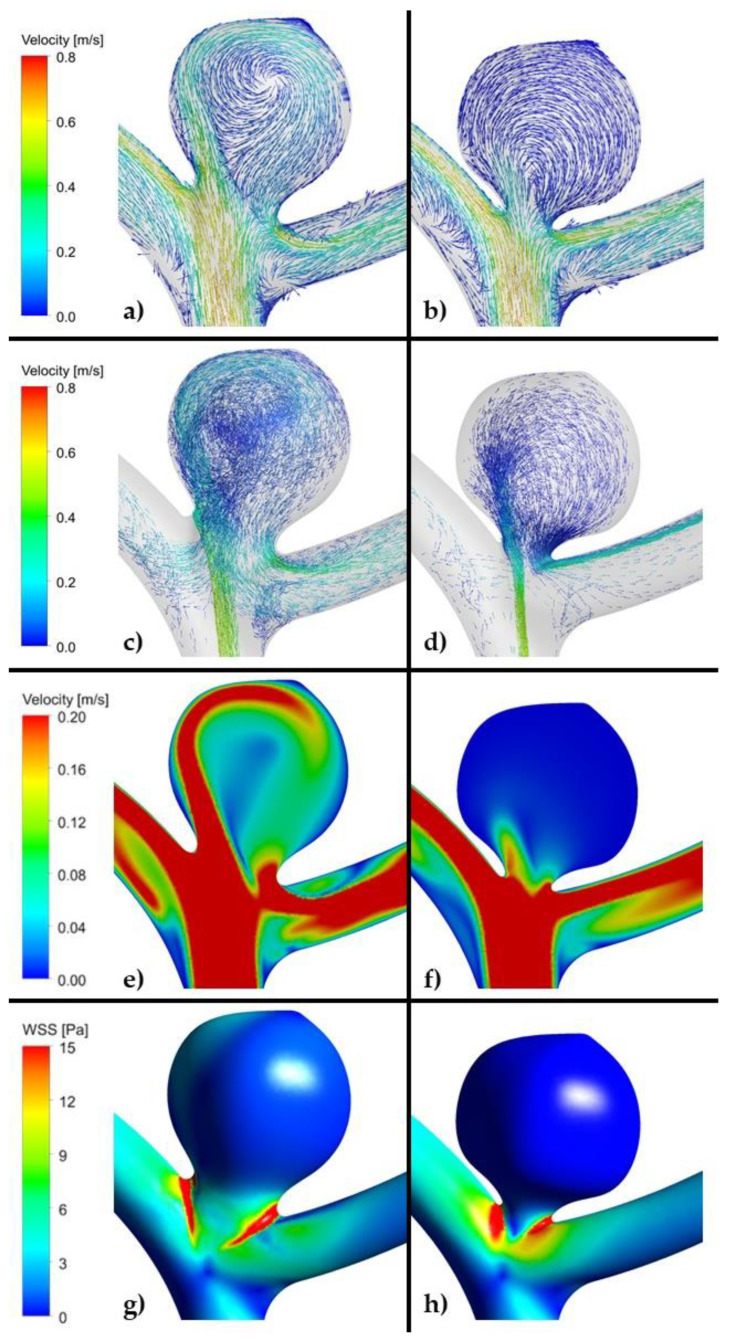
(**a**,**c**,**e**,**g**). Example of high risk recanalization aneurysm model (unruptured group). Flow pattern analysis demonstrates a complex vortex structure. WSS analysis demonstrates heterogenous wall shear stress with an area of the aneurysm dome under high shear stress. (**b**,**d**,**f**,**h**). Example of low risk recanalization aneurysm model (unruptured group). Flow pattern analysis demonstrates a simple vortex structure. WSS analysis demonstrates heterogenous wall shear stress with an area of the aneurysm dome under low shear stress. (ANSYS Inc. Post-processing module, Canonsburg, PA, USA).

**Figure 6 jpm-11-00793-f006:**
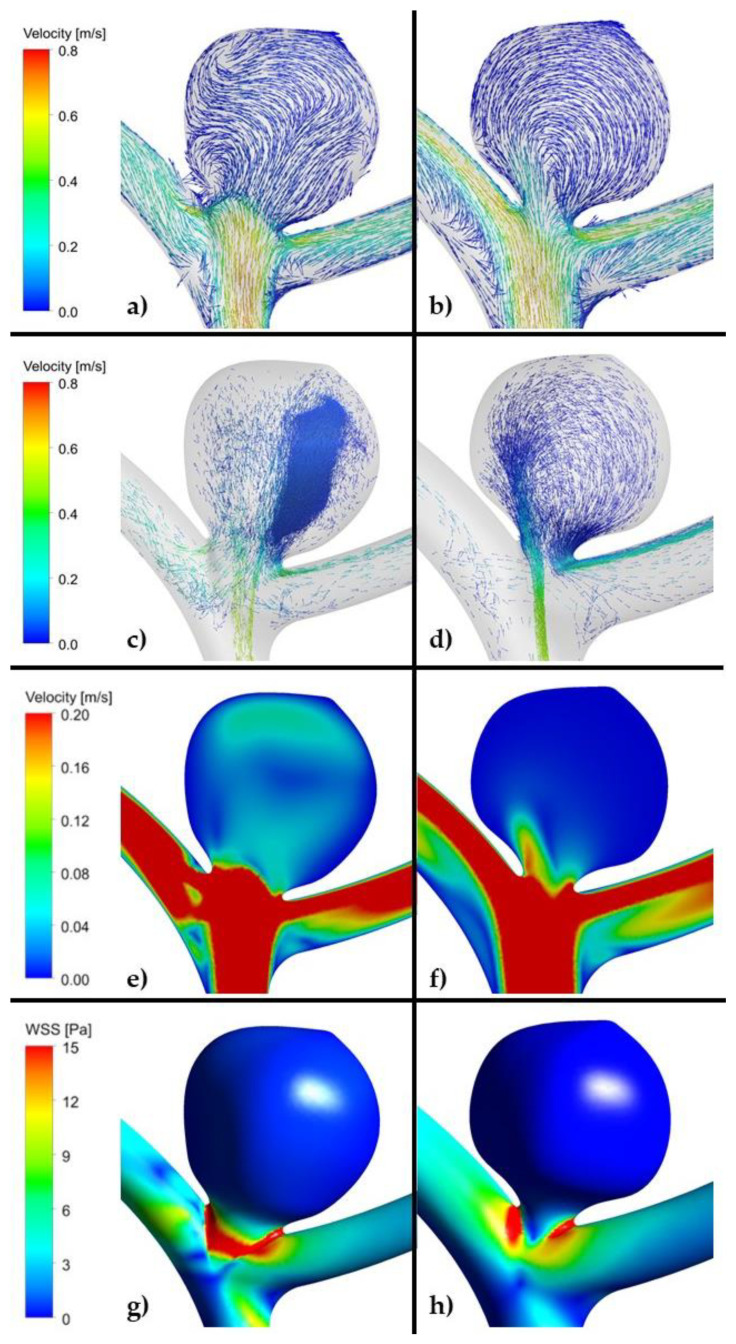
(**a**,**c**,**e**,**g**). Example of high risk recanalization aneurysm model (ruptured group). Flow pattern analysis demonstrates a complex vortex structure. WSS analysis demonstrates heterogenous wall shear stress with an area of the aneurysm dome under high shear stress. (**b**,**d**,**f**,**h**). Example of low risk recanalization aneurysm model (ruptured group). Flow pattern analysis demonstrates a simple vortex structure. WSS analysis demonstrates heterogenous wall shear stress with an area of the aneurysm dome under low shear stress. (ANSYS Inc. Post-processing module, Canonsburg, PA, USA).

**Table 1 jpm-11-00793-t001:** Inclusion and exclusion criteria.

	Study Group
Inclusion criteria	Single saccular aneurysm treated with classical embolization without the use of stent and balloon remodeling. Full medical documentation of the hospitalization period and the follow-up visits with DSA done, 6 months after the treatment.
Exclusion criteria	Admission to hospital later than 30 days after aSAH. History of prior intervention (endovascular or microsurgical) of the treated aneurysm.

**Table 2 jpm-11-00793-t002:** Characteristic parameters.

**Unruptured Aneurysm Group**			
**Value**	**Mean ± SD without Recanalization**	**Mean ± SD with Recanalization**	***p***
age (years)	54.1 ± 11.6	56.5 ± 14.0	0.350
depth (mm)	6.7 ± 3.6	12.0 ± 6.6	<0.001 *
height (mm)	8.2 ± 4.7	13.8 ± 6.6	<0.001 *
width (mm)	6.6 ± 3.8	12.1 ± 6.8	<0.001 *
neck size (mm)	3.6 ± 1.0	5.0 ± 2.3	<0.001 *
APTT (s)	29.4 ± 6.2	29.1 ± 3.4	0.806
INR	1.0 ± 0.1	1.0 ± 0.1	0.849
HCT (%)	38.4 ± 9.3	38.9 ± 7.9	0.764
prothrombin time (s)	101.3 ± 6.7	98.2 ± 6.5	0.022
aneurysm volume (mm^3^)	426.7 ± 1287.4	1794.1 ± 2202.0	<0.001 *
packing density	33.3 ± 8.0%	20.0 ± 8.9%	<0.001 *
parent artery diameter (mm)	3.8 ± 0.9	4.3 ± 1.1	0.012
maximum aneurysm height/parent artery diameter—SR ratio	2.3 ± 1.3	3.4 ± 1.6	<0.001 *
neck size/parent artery diameter	1.0 ± 0.3	1.2 ± 0.5	0.004
the maximal perpendicular height (mm)	7.7 ± 4.3	12.9 ± 6.0	<0.001 *
the maximal perpendicular height/neck size (aspect ratio)	7.8 ± 4.6	12.7 ± 6.0	<0.001 *
vessel angle (degree)	60.7 ± 24.1	60.4 ± 22	0.943
aneurysm angle (degree)	85.1 ± 19.5	82.0 ± 20.8	0.455
aneurysm depth/neck size (AR ratio)	7.9 ± 1.0	11.8 ± 6.3	0.039
Complete aneurysm filling during the first embolization (*p* < 0.001 *)			
**Modified Raymond-Roy Scale Class**	**(N)**		
I	107		
II	7		
IIIa	2		
IIIb	2		
**Ruptured Aneurysm Group**			
**Value**	**Mean ± SD without recanalization**	**Mean ± SD with recanalization**	***p***
age (years)	56.7 ± 15.3	56.9 ± 16.1	0.974
depth (mm)	5.5 ± 2.7	6.4 ± 3.3	0.229
height (mm)	6.9 ± 3.2	9.5 ± 5.2	0.016
width (mm)	5.4 ± 2.6	6.6 ± 3.3	0.111
neck size (mm)	3.3 ± 0.8	4.1 ± 0.7	0.001 *
APTT (s)	30.1 ± 5.5	29.0 ± 2.7	0.473
INR	1.1 ± 0.3	1.1 ± 0.1	0.916
HCT (%)	35.5 ± 11.4	36.5 ± 10.1	0.759
prothrombin time (s)	1392.4 ± 6314.5	95.2 ± 8.3	0.389
aneurysm volume (mm^3^)	166.5 ± 274.3	349.1 ± 432.4	0.045
packing density	35.0 ± 10.8%,	21.2 ± 6.6%	0.001 *
parent artery diameter (mm)	3.7 ± 0.9	3.9 ± 1.1	0.525
maximum aneurysm height/parent artery diameter—SR ratio	2.0 ± 1.1	2.5 ± 1.3	0.147
neck size/parent artery diameter	0.9 ± 0.3	1.2 ± 0.5	0.015
the maximal perpendicular height (mm)	6.9 ± 3.4	12.0 ± 5.1	0.001 *
the maximal perpendicular height/neck size (aspect ratio)	2.1 ± 0.9	3.0 ± 1.6	0.006
vessel angle (degree)	62.5 ± 26.0	62.0 ± 22.2	0.934
aneurysm angle (degree)	85.3 ± 22.7	90.8 ± 26.9	0.402
aneurysm depth/neck size (AR ratio)	1.7 ± 0.8	1.6 ± 0.9	0.590
Complete aneurysm filling during the first embolization (*p* < 0.001 *)			
**Modified Raymond-Roy Scale Class**	**(N)**		
I	52		
II	11		
IIIa	1		
IIIb	2		

*p* values calculated using *t*-test; values marked with * remained significant after correction for multiple comparisons.

**Table 3 jpm-11-00793-t003:** Aneurysm location.

Ruptured Aneurysms (*n* = 66); Anterior Part of the Circle of Willis (*n* = 42, 63.6%), Posterior Part (*n* = 24, 36.4%)	Unruptured Aneurysms (*n* = 118); Anterior Part of the Circle of Willis (*n* = 84, 71.2%), Posterior Part (*n* = 34, 28.8%)
ACoA (*n* = 13, 19.7%)	ACoA (*n* = 6, 5.1%)
MCA *n* = 15, 22.7% (M2 *n* = 14), (M1 *n* = 1)	MCA *n* = 6, 5.1% (M2 *n* = 3), (M1 *n* = 3)
ICA (*n* = 12, 18.2%): PCoA (*n* = 6; include 2 fetal type PcoA) ICA bifurcation (at the apex *n* = 2) ophthalmic segment (originating at the ophthalmic artery *n* = 1) anterior choroidal artery (*n* = 1)	ICA (*n* = 69, 58.5%): PCoA (*n* = 6; include 1 fetal type PcoA) ICA bifurcation aneurysms at apex (*n* = 16) and those projecting anteriorly (*n* = 7) and dorsally (*n* = 4); (*n* = 27) ophthalmic segment (originating at the ophthalmic artery (*n* = 26) and at the superior hypophyseal artery (*n* = 6); (*n* = 32) anterior choroidal artery (*n* = 4)
PCallA (*n* = 2, 3.0%)	PCallA (*n* = 3, 2.5%)
BA (*n* = 17, 25.8%); above (*n* = 7), at (*n* = 5) and below the level of the posterior clinoid (*n* = 5).	BA (*n* = 21, 17.8%); above (*n* = 9), at (*n* = 8) and below the level of the posterior clinoid (*n* = 4).
PICA (*n* = 2, 3.0%)	PICA (*n* = 2, 1.7%)
PCA (*n* = 1, 1.5%)	AICA (*n* = 2, 1.7%)
SCA (*n* = 1, 1.5%)	SCA (*n* = 4, 3.4%)
VA (*n* = 3, 4.5%); 2 aneurysms arising from proximal carina of fenestration at VB junction (in 1 case it involved both limbs) and 1 originating along the entire limb.	VA (*n* = 5, 4.2%); 3 aneurysms arising from proximal carina of fenestration at VB junction (in 2 cases it involved both limbs) and 2 originating along the entire limb.

ACoA—anterior communicating artery; MCA—middle cerebral artery (M1—proximal part of MCA, M2 bifurcation part of MCA); CA—internal carotid artery; PCoA—posterior communicating artery; PCallA—pericallosal artery; BA—basilar artery; PCA—posterior artery; PICA—posterior inferior cerebellar; AICA—anterior inferior cerebellar; SCA—superior cerebellar artery; VA—vertebral artery; percentages may not add up to 100% due to rounding.

**Table 4 jpm-11-00793-t004:** Multivariable analysis results.

Recanalization after 6 Months	Odds Ratio (95%CI)	*p* Value
aSAH group		
the degree of aneurysm filling	17.00 (2.32–124.40)	0.005
during the first embolization		
packing density	0.87 (0.79–0.97)	0.01
Unruptured group		
packing density	0.84 (0.79–0.91)	<0.001
neck size (mm)	2.16 (1.23–3.80)	0.008
Recanalization after 12 months	Odds Ratio (95%CI)	*p* value
aSAH group		
the degree of aneurysm filling	6.88 (0.98–48.29)	0.052
during the first embolization		
packing density	0.85 (0.77–0.94)	0.002
Unruptured group		
packing density	0.84 (0.78–0.91)	<0.001
neck size (mm)	5.46 (2.52–11.83)	<0.001

**Table 5 jpm-11-00793-t005:** Hemodynamic parameters.

		Reference	Case1		Case2		Case3		Case4		Case5		Case6		Case7	
			High Risk	Low Risk	High Risk	Low Risk	High Risk	Low Risk	High Risk	Low Risk	High Risk	Low Risk	High Risk	Low Risk	High Risk	Low Risk
Unruptured aneurysms group	max WSS (Pa)	97.927	93.084	97.481	90.476	96.204	85.593	93.084	79.645	93.084	90.776	95.748	92.150	93.084	90.411	95.748
	areaAve WSS (Pa)	2.538	2.423	3.071	1.306	2.024	3.731	2.423	2.846	2.423	1.001	1.373	2.316	2.423	1.154	1.373
	areaAve OSI (–)	0.160	0.211	0.200	0.210	0.201	0.163	0.211	0.237	0.211	0.198	0.198	0.156	0.211	0.195	0.198
	max TAWSS (Pa)	43.779	41.708	43.070	40.477	42.689	31.265	41.708	37.525	41.708	39.932	42.196	36.392	41.708	40.287	42.196
	areaAve TAWSS (Pa)	1.242	1.021	1.305	0.570	0.880	1.646	1.021	1.246	1.021	0.432	0.591	0.977	1.021	0.491	0.591
Ruptured aneurysms group	max WSS (Pa)	123.535	95.748	102.925	101.551	132.148	93.084	102.925	92.150	93.084	-	-	-	-	-	-
	areaAve WSS (Pa)	1.659	1.373	2.774	2.692	2.597	2.423	2.774	2.316	2.423	-	-	-	-	-	-
	areaAve OSI (–)	0.207	0.198	0.231	0.157	0.209	0.211	0.231	0.156	0.211	-	-	-	-	-	-
	max TAWSS (Pa)	54.162	42.196	46.33	45.18	53.879	41.708	46.33	36.392	41.708	-	-	-	-	-	-
	areaAve TAWSS (Pa)	0.746	0.591	1.179	1.223	1.047	1.021	1.179	0.977	1.021	-	-	-	-	-	-

**Table 6 jpm-11-00793-t006:** Area-averaged velocity at aneurysm neck cross-section in each investigated case study. # case study.

Values That Increase Recanalization Risk, Based on ROC Curve Analysis		Case	Blood Velocity at Aneurysm Neck (m/s); Measurements Taken at Systole Peak		
			Reference	High Risk of Recanalization	Low Risk of Recanalization
Unruptured group	S > 9.3 mm	#1	0.167	0.104 (−38%)	0.103 (−38%)
	H > 13 mm	#2		0.110 (−34%)	0.107 (−36%)
	N > 4 mm	#3		0.156 (−7%)	0.104 (−38%)
	P > 4.6 mm	#4		0.124 (−26%)	0.104 (−38%)
	SR ratio (H/N ratio) > 2.759	#5		0.104 (−38%)	0.105 (−37%)
	N/P ratio > 1.042	#6		0.163 (−2%)	0.104 (−38%)
	aspect ratio (H/N ratio) > 10.526	#7		0.116 (−31%)	0.105 (−37%)
Ruptured group	H > 12 mm	#1	0.119	0.105 (−12%)	0.106 (−11%)
	N > 3.6 mm	#2		0.137 (−15%)	0.095 (−20%)
	aspect ratio (H/N ratio) > 3.075	#3		0.104 (−13%)	0.106 (−11%)
	N/P ratio > 1.023	#4		0.163 (37%)	0.104 (−13%)

**Table 7 jpm-11-00793-t007:** Recanalization risk factors; literature review.

Major Studies of Intracranial Aneurysms. Number of Patients Included, Aneurysm Status (Ruptured or Unruptured) and Recanalization Risk Factors					
			Risk Factors		
Study	Patients	IA Status	Patient Factors	Aneurysm Factors	Treatment Factors
CLARITY trial [60]	*n* = 782	ruptured	age < 65 y.o.	neck size ≥ 4 mm	-
Ortiz et al. [61]	*n* = 110	ruptured/unruptured	cigarette smoking	-	-
Raymond et al. [10]	*n* = 446	ruptured/unruptured	-	ruptured status/aneurysms greater than 10 mm/neck size ≥ 4 mm	incomplete occlusion immediately after procedure
Hope et al. [62]	*n* = 58	ruptured/unruptured	-	ruptured status/neck size ≥ 4 mm	incomplete occlusion immediately after procedure
Plowman et al. [63]	*n* = 570	ruptured	-	ruptured status	-
Ferns et al. [64]	*n* = 400	ruptured/unruptured	-	location at basilar tip	-
Ries et al. [65]	*n* = 323	ruptured/unruptured	-	aneurysms greater than 10 mm/neck size ≥ 4 mm	-
Choi et al. [66]	*n* = 87	ruptured/unruptured	-	neck size ≥ 4 mm	-
Piotin et al. [67]	*n* = 223	ruptured/un ruptured	-	-	incomplete occlusion immediately after procedure
Grunwald et al. [9]	*n* = 211	ruptured/unruptured	-	-	incomplete occlusion immediately after procedure
Sluzewski et al. [58]	*n* = 198	ruptured/unruptured	-	-	packing density less than 24%
Liu et al. [50]	*n* = 200	ruptured/unruptured	-	ruptured status/neck size ≥ 4 mm/those aneurysms with atherosclerotic plaques	-
Jeon et al. [48]	*n*= 898	ruptured/unruptured	-	-	Stent deployment significantly decreased recanalization rate
ISAT study [17]	*n* = 2143	ruptured	the patient’s young age	aneurysms greater than 10 mm	incomplete occlusion immediately after procedure
Brillstra et al. [51]	*n* = 1383	ruptured/unruptured	-	neck size ≥ 4 mm	-
Hayakawa et al. [52]	*n* = 71	ruptured/unruptured	-	aneurysms greater than 10 mm	-

## Data Availability

The data presented in this study are openly available in open research data repository (Zenondo).

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
