# Peer review of "Risk Factors for Recanalization after Coil Embolization"

_jpm, 2021, doi:10.3390/jpm11080793_

Round 1
Reviewer 1 Report
This is an outstanding article authors should be congratulated for. All my suggestions have been answered or considered.
Reviewer 2 Report
The authors have fulfilled all the requested remarks from my review. Addition of supplementary material has enriched the publication.
This manuscript is a resubmission of an earlier submission. The following is a list of the peer review reports and author responses from that submission.
Round 1
Reviewer 1 Report
This is a retrospective study which seeks to assess morphometric, clinical and dynamic risk factors of recanalization in aneurysms treated by coiling.
Authors should be commended for their efforts on analysing such a concerning issue.
Introduction might be too long, and the precise scope of this manuscript is loosely described considering too general questions regarding aneurysms natural history.
Methods are extensively described and results are clearly presented.
As authors note, regarding CDF analysis this is an interim analysis and, therefore, a small paragraph describing future perspectives and next steps to be done in this field should be included on the discussion.
Thus, intrasaccular flow pattern can be treated as a relevant recanalization predictor, especially knowing that this parameter is responsible for coil mesh compaction. This statement needs internal/external validation.
I consider that the strenght of this article is de CFD analysis, therefore, discussion should be focused on this and not on other well known factors of recanalization.
Authors may consider commenting on the posibility of building 3D CFD models based on diagnostic DSA and the potential benefits of implementing DSA instead of CTA.
Reviewer 2 Report
Dear Authors,
Needless to say, aneurysm sac recanalization is a challenge in everyday interventional radiology practice and more high quality research is needed to understand and combat this phenomenon. I appreciate the amount of work you obviously put into this article, however there are number of substantial issues afflicting it.
1) the article needs thorough language revision, the introduction resembles the discussion part of the article, and needs to be rewritten
2) recenalization rates reports vary substantially, with largest series reporting 3-5% in 6mths to 5 years follow-up, these are not mentioned in your discussion
3) you state re-embolization or surgical clipping... carries a risk of aSAH, while you provide no reference to back this statement. The literature indicates differently
4) 410cases in 8 years corresponds to around 4 cases/month which is of low-volume center, which might affect recanalization rates
5) measurements were depicted for vessel division region, were sidewall aneurysms included? branches originating for the aneurysm sac or neck should also be taken into account in the calculations
6) what was the rationale for described exclusion criteria?
7) none of the described risk factors were not already included in previous reports, what would you consider a novelty in your analysis?
Reviewer 3 Report
The authors present a retrospective study on 184 patients who underwent coiling for cerebral aneurysms in order to identify risk factors for coil recanalization. Two models were created with high and low risk profile for recanalization based on ROC curves with CDF for validation of results.aneurysn filling during the first embolization was independent risk factor and packing density was protective factor for recanalization in ruptured and unruptured aneurysms with neck size as an independent risk factor.
AAlthough authors present their results in fair and understandable way it is unclear which consequences for follow up and treatment the high risk group has. The study is underpowered because of mix of ruptured and unruptured aneurysms, low number of patients and retrospective character. Complex flow and vortices were as expected associated with the shape and were associated with high risk. Please provide literature review in Table form with the most important studies number of patients and risk factors
Please comment extensively on value of anticoagulation for prevention and describe your standard. While authors provide a solid hemodynamic study, it would be important at least to identify smoking, diabetes and hypertension as risk factors for study to be completed; this should be done for all 184 patients. Was the velocity on neck plane measured, or wall shear and did it decline in the recanalization group.